# Protective Effects of Dietary Resveratrol against Chronic Low-Grade Inflammation Mediated through the Gut Microbiota in High-Fat Diet Mice

**DOI:** 10.3390/nu14101994

**Published:** 2022-05-10

**Authors:** Pan Wang, Yue Ma, Dan Wang, Wenting Zhao, Xiaosong Hu, Fang Chen, Xiaoyan Zhao

**Affiliations:** 1Beijing Key Laboratory of Agricultural Products of Fruits and Vegetables Preservation and Processing, Key Laboratory of Vegetable Postharvest Processing, Ministry of Agriculture and Rural Affairs, Institute of Agri-Food Processing and Nutrition, Beijing Academy of Agriculture and Forestry Sciences, Beijing 100097, China; wp_6767@126.com (P.W.); mayue@iapn.org.cn (Y.M.); wangdan@iapn.org.cn (D.W.); zhaowenting@iapn.org.cn (W.Z.); 2Key Laboratory of Fruits and Vegetables Processing, Ministry of Agriculture, Engineering Research Centre for Fruits and Vegetables Processing, National Engineering Research Center for Fruit and Vegetable Processing, College of Food Science and Nutritional Engineering, China Agricultural University, Beijing 100083, China; huxiaos@263.net

**Keywords:** resveratrol, inflammation, gut microbiota, fecal microbiota transplantation

## Abstract

Resveratrol (RSV), a natural polyphenol, has been shown to exert activity against obesity and related chronic inflammation. However, due to the poor bioavailability of RSV, the mechanisms of RSV against inflammation in obesity models remain unclear. In this study, we aimed to investigate the relationship between the gut bacteria and the anti-inflammation effects of RSV in HFD-fed mice. We found that RSV supplementation reduced fat accumulation and improved systemic inflammation in HFD-fed mice. Meanwhile, RSV attenuated HFD-induced changes in the gut microbiota’s structure, which were associated with inflammatory parameters. A fecal microbiota transplantation (FMT) experiment proved that the anti-inflammation effects of RSV largely rely on the gut microbiota. Moreover, the microbiota-genera-changing trend in the FMT experiment was similar to that in the oral RSV-feeding experiment. Thus, these results demonstrate that modulation of the gut bacteria induced by RSV treatment has a therapeutic effect on chronic low-grade inflammation in HFD-fed mice.

## 1. Introduction

Obesity and overweight have become global epidemics [1]. Excessive body fat accumulation is the hallmark of obesity and is associated with chronic, low-grade, systemic inflammation. Abnormal adipocytes and hepatocytes secrete proinflammatory cytokines, such as TNF-α, IL-1β and IL-6 [2]. Meanwhile, low-grade inflammation leads to macrophage recruitment [3], which stimulates glucose and fatty acid metabolism disorders and induces insulin resistance, obesity and other chronic metabolic diseases. Therefore, it is necessary to investigate safe and effective strategies for reducing chronic low-grade inflammation.

Gut bacteria are a crucial environmental factor for the development of obesity and systematic inflammation. Trillions of bacteria are colonized densely in the gut, which contribute to nutrient acquisition and energy regulation [4]. Consumption of a high-fat diet (HFD) could induce gut microbiota dysbiosis, such as an increased ratio of *Firmicutes* to *Bacteroidetes* [5]. Intriguingly, transplanting HFD-induced gut bacteria to germ-free mice increased their body weight and caused complications of metabolic syndrome [6]. Moreover, a HFD also contributes to the higher levels of intestinal Gram-negative bacteria-derived lipopolysaccharide (LPS) in the systemic circulation [7]. The LPS plays a key role in the progression of inflammation that ultimately leads to insulin resistance and related metabolic diseases [8]. Moreover, high LPS in mice models leads to adipose inflammation and insulin resistance [9]. Hence, the gut microbiota is a crucial target for dietary intervention against inflammation and obesity [10,11,12].

Resveratrol (RSV), is a natural bioactive compound with a plethora of beneficial effects, including anti-inflammatory, antioxidant, anti-hyperlipidemic and anti-obesity [13]. However, the beneficial effects of RSV are limited due to its low bioavailability and incomplete absorption in the intestine. Intriguingly, RSV exhibits high accumulation and persists for a long time in the intestinal tissue after oral administration [14]. This indicates that the gut tissue is the key site for RSV’s beneficial effects. Moreover, recent evidence has shown that RSV could improve intestinal microecology in obese individuals, including increasing the relative abundances of *Bacteroides* and *Blautia* and decreasing the relative abundances of *Akkermansia, Lachnospiraceae Moryella* and *Turicibacteraceaea* [15,16]. In our previous study, we provided evidence that the interaction between RSV and gut microbiota plays a key role in controlling obesity development [13]. Additionally, systematic inflammation has long been regarded as a contributing factor to obesity. Nevertheless, it remains unclear whether RSV-induced alleviation of systematic inflammatory status is mediated by the gut microbiota.

In this study, we aimed to determine whether the anti-inflammation effect of RSV administration is regulated by the gut bacteria. We investigated the influence of RSV on systematic inflammation in HFD-fed mice. Then, 16S rRNA analysis was used to determine the effect of RSV on mouse gut bacteria composition. Additionally, the relationships between the intestinal microenvironment and systematic inflammation parameters were also explored. Moreover, a fecal microbiota transplantation (FMT) experiment was used to discover whether the gut bacteria are causally involved in the anti-inflammation associated with RSV supplementation.

## 2. Materials and Methods

### 2.1. Animal Experiment

The animal studies were approved by the Biomedical Ethical Committees of Peking University (Beijing, China) (approval number LA2018288). The experiment protocols were confirmed by the Animal Care Committee of Peking University and corresponded with the Guide for the Care and Use of Laboratory Animals (National Institutes of Health (NIH), Bethesda, MD, USA).

Male Five-week-old C57BL/6 J mice were housed in a standard specific-pathogen-free (SPF) facility at 25 ± 2 °C (4 mice/cage, 12-h day/night cycle). With 1 week of adaptation on, mice were split into 3 groups randomly. The experiment lasted for 16 weeks. One group was fed a standard diet (SD, containing10% fat by energy, *n* = 12) group; the other two groups were fed a high-fat diet (HFD, containing 60% fat by energy, *n* = 12) or an identical diet treated with RSV (300 mg/kg/day) by gavage (HFDR, *n* = 12). Related ingredients and energy densities of these diets are shown in Appendix A. Diet intake, body weight and body weight gain were measured every week. At the end of the study, stool samples were collected and kept at −80 °C. Moreover, animals were deprived of food for 12 h and euthanized at the end of 16 weeks. Blood was drawn and centrifuged immediately (3000 rpm, 10 min). Tissues were collected, weighted and stored at −80 °C after 12 h of fasting.

### 2.2. Fecal Microbiota Transplantation (FMT)

Six samples were chosen randomly from HFD (*n* = 6) and HFDR (*n* = 6) groups to act as donor mice. Fresh stool samples were collected daily for the fecal microbiota transplantation (FMT) experiment. Stools from donor mice of each group were pooled, and 200 mg was suspended in 5 mL of PBS. The solution was vortexed for 3 min, and centrifugation followed at 800× *g* for 3–5 min. Five-week C57BL/6J male recipient mice (*n* = 6 in each transplant group) were fed with a HFD. After 1 week of acclimation, fresh mix antibiotics (ampicillin 1 g/L, vancomycin 500 mg/L, neomycin 1 g/L, metronidazole 1 g/L) were added into the drinking water of the recipient mice for 2 weeks. After antibiotic treatment, the water was replaced with regular water. Then, the recipient mice were gavaged with the microbiota of donor mice from the HFD and HFR groups, via 200 μL fresh supernatant daily from the donors. After 16 weeks treatment, animals were euthanatized and tissues were collected.

### 2.3. Biochemical Analysis

Content of serum LPS was quantified by a commercial kit (Xiamen Limulus Ex-perimental Reagents Factor, Xiamen, China). The concentrations of glucose (GLU), triglyceride (TG), total cholesterol (TC), low-density lipoprotein (LDL), high-density lipoprotein (HDL), alanine aminotransferase (ALT) and aspartate aminotransferase (AST) in serum were measured using an automatic biochemistry analyzer (Hitachi Ltd., Tokyo, Japan). The levels of IL-6 and TNF-α in serum were detected by commercial ELISA kits (Nanjing Jiancheng Bio-engineering Institute, Nanjing, China).

### 2.4. Fluorescein Isothiovyanate-Dextran Permeability Test

According to previous study, the method was used to assess intestine epithelial barrier permeability in vivo with minor modification [17]. At week 14, mice were fasted for 4 h before administration with FITC–dextran (Sig-ma-Aldrich, St. Louis, MO, USA) by gavage. The FITC–dextran was dissolved at a dose of 0.8 mg/kg. After 4 h later, collected the serum, diluted with PBS (1:1), and analyzed on a plate reader (excitation wavelength of 485 nm, emission wavelength of 535 nm).

### 2.5. Gene Expression Analysis

Six samples were chosen randomly from each group for the gene expression analysis. Total mRNA from different tissue was isolated by a Trizol reagent (Invitrogen, Waltham, MA, USA) and purified with an RNeasy Mini Kit (Qiagen, Venlo, The Netherlands). qRT-PCR analysis was performed using the LightCycler 480 Real-Time PCR system (Roche Diagnostics, Basel, Switzerland) with the SYBR Green. The primers used in this study are listed in Appendix A. The relative expression levels of these target genes were normalized to GAPDH and calculated by the 2^−ΔΔCt^ method.

### 2.6. Histological Analysis

Epididymal white and interscapular brown adipose tissues fixed in 4% paraformaldehyde were embedded in paraffin and sections (5 μm). The slices were stained with hematoxylin and eosin (HE staining).

### 2.7. 16S rRNA Sequencing

Six samples were chosen randomly from each group for the 16S rRNA sequencing in the experiment mentioned in Section 2.1. These six mice were also the donor mice in the FMT experiment. Genomic DNA was extracted from 0.5 g fecal samples with a DNA stool mini kit (Qiagen, Beijing, China; *n* = 6). The V3–V4 hypervariable regions of the 16S rRNA were amplified by using primers 338F and 806R, then sequenced by Illumina MiSeq (Illumina, Shanghai, China), as described in a previous study [13]. High quality reads were performed for bioinformatics analysis and the operational taxonomic units (OTUs) were identified based on a 97% similarity level (Supplementary method).

### 2.8. Statistical Analysis

The investigators were blinded to the group allocation during the experiment. Sample size for each experiment was determined on the basis of previous reports using mouse models of obesity. All data are presented as means ± SEM. Statistical significance analyses were evaluated by GraphPad Prism version 6.0 (GraphPad Software Inc., La Jolla, CA, USA). The results from two groups were analyzed with unpaired Student’s t tests. The results of more than two groups were analyzed with one-way ANOVA followed by Tukey’s post hoc test. *p* values < 0.05 were considered as statistically significant. Correlations of gut bacteria and inflammation parameters were based on Spearman’s rank-order correlation calculations.

## 3. Results

### 3.1. Dietary RSV Reduced Fat Accumulation and Systemic Low-Grade Inflammation in HFD-Fed Mice and HFD-Induced Metabolic Endotoxemia Plays a Causal Role in the Development of Inflammation and Related Metabolic Diseases

As expected, the HFD group had greater body weight, body weight gain, total fat accumulation and adipocyte size compared to the SD group. RSV administration significantly reduced body weight and fat accumulation in HFD-fed mice (Figure 1A,B, Appendix A). Nevertheless, no significant difference was found in mean energy intake or energy efficiency between HFD-fed groups. In addition, RSV efficiently decreased HFD-induced elevations in the serum levels of glucose, insulin, TC, TG and HDL-C, and liver levels of AST and AST (Appendix A).

Moreover, RSV supplementation significantly suppressed the absorption of FITC–dextran and LPS levels in the serum in the HFD mice, which indicated an improved intestinal barrier (Figure 1C,D). Circulating LPS causes systemic low-grade inflammation, so we further assessed the impact of RSV administration on systematic inflammation. RSV supplementation significantly decreased the concentrations of serum TNF-1α and IL-6 in HFD-fed mice (Figure 1E,F). Consistently with previous studies, we also found that HFD induced a pro-inflammatory effect on the jejunum, adipose and liver tissue. We measured the mRNA expression of pro-inflammatory cytokines in these tissues. The expression levels of TNF-1α and IL-6 were higher in the jejunum, adipose and liver tissue of the HFD group than the SD group, but RSV treatment reduced the expression of these cytokines (Figure 1G,H). In addition, RSV treatment reduced the expression of IL-1 in the jejunum and adipose tissue of HFD-fed mice (Figure 1I). MCP-1 promotes macrophage accumulation, and RSV treatment also decreased the expression of MCP-1 in the jejunum and white adipose tissue of HFD-fed mice (Figure 1J). These results proved that RSV treatment alleviated HFD-induced fat accumulation and systemic inflammation.

### 3.2. Dietary RSV Improved the Gut Microbiota Structure in HFD-Fed Mice

To better understand whether the beneficial anti-obesity and anti-inflammation effects of RSV supplementation in HFD-fed mice were due to the alteration of gut bacteria composition, stool samples were collected for the further analysis of mice in the three groups. In total, 1,807,893 raw reads were detected from 18 samples. After selecting the effective reads, 1,508,039 effective reads were generated, and each fecal sample produced an average of 83,780 clean sequences per sample that were used for the downstream analysis. For alpha diversity analysis, ACE, Chao, Simpson and Shannon indexes were used to estimate community richness and diversity. There was no statistical difference in species richness (Chao 1 and Ace index) among SD, HFD and HFDR groups (Appendix A). The species diversity (Shannon) was higher in the HFDR group than in the other two groups (Appendix A). The Simpson diversity of the HFDR group was slightly lower than that of the HFD group (Appendix A). These results proved that HFD downregulated the species diversity of gut microbiota, and supplementation with RSV could elevate the diversity. The results of NMDS and ANOSIM analysis suggested significantly different beta diversity between these groups (Figure 2A and Appendix A).

In our previous study, we found that at the phyla level, RSV treatment did not reverse the dominant bacteria in three groups, such as *Firmicutes* and *Bacteroidetes* [13]. To further evaluate the ecology changes of the gut bacterial induced by RSV, we analyzed the relative abundances at different classification levels among the three groups. At the class level, RSV increased the abundance of *Erysipelotrichia* compared to the HFD group, and the abundances of *Clostridia* tended to decrease (*p* = 0.08) with RSV feeding in HFD-fed mice (Appendix A). At the order level, RSV treatment tended to increase the abundance of *Erysipelotrichales* and decrease the abundance of *Clostridiales* (*p* = 0.08) in HFD-fed mice (Appendix A). At the family level, RSV supplementation significantly upregulated the levels of *Erysipelotrichaceae* and *Bacteroidaceae*, but downregulated the levels of *Bacteroidales_S24-7_group, Desulfovibrionaceae,* and *Ruminococcaceae* in HFD-fed mice (Appendix A). At the genus level, we analyzed the top 10 genera and found that the levels of *Allobaculum*, *Bacteroides* and *Blautia* were increased after RSV supplementation in the HFD-fed mice [13]. The levels of *Desulfovibrio* and *Lachnospiraceae_NK4A136_group* were decreased in the HFDR group compared to the HFD group [13] (Figure 2B,C). We aimed to explore some new genera with low abundances with significant differences between HFD and HFDR groups. We analyzed the top 35 genera and found that the abundance of *Parabacteroides* was increased in the HFDR group. The levels of *Anaerotruncus, Oscillibacter, Romboutsis, Lachnospiraceae_UCG-006, Coprococcus1* and *Roseburia* were decreased after RSV treatment in HFD-fed mice (Figure 2D).

### 3.3. Gut Microbiota Involved in the Systematic-Inflammation-Alleviating Effects of RSV

To investigate whether the altered gut microbiota was related to the alleviated systematic inflammation after RSV treatment, Spearman’s rank correlation test was used to evaluate the correlations between the 10 differentially abundant genera and systematic inflammation parameters. The results suggest that *Desulfovibrio*, *Lachnospiraceae_NK4A136_group*, *Akkermansia*, *Rikenellaceae_RC9_gut_group* and *Anaerotruncus* were significantly positively correlated with at least one systematic inflammation parameter; and *Allobaculum*, *Bacteroides*, *Blautia*, *Lachnoclostridium* and *Parabacteroides* were negatively correlated with at least one systematic inflammation parameter (Figure 3).

### 3.4. Effect of FMT on Systematic Inflammation and the Gut Microbiota’s Structure

To further elucidate the role of the gut bacterial in regulating the protective effects of RSV on systematic inflammation, we transferred the bacteria from HFD and HFDR groups to recipient mice fed with HFD, followed by measuring the systematic inflammation related traits (Figure 4A). FMT from HFDR-treated mice reduced body weight and obesity trait in HFD recipients (Appendix A). Moreover, the serum biochemical parameters were almost improved by the HFDR microbiota (Appendix A). In addition, there was a remarkable improvement in systematic inflammation, including the level of the serum TNF-1α and IL-6 (Figure 4B), and the most mRNA expression of pro-inflammatory cytokines in tissues (Figure 4C–F).

To confirm that fecal transplantation modulated the gut ecology, we examined the structure of gut microbiota using 16S rRNA. Totally, 2,005,839 raw reads were detected, and after selecting the effective reads, 1,673,569 effective reads were generated (Appendix A). The Shannon indexes were dramatically elevated in the HFDR→HFD group compared to the HFD→HFD group. The other three indexes (Shannon, Ace and Chao) showed no significant difference between the HFD→HFD group and the HFDR→HFD group (Appendix A). The results of NMDS and ANOSIM analysis suggested significantly different beta diversity among these groups. (Figure 5A and Appendix A). From the heatmap comparison and hierarchical clustering dendrogram based on the relative abundances of top the 35 genera across the four groups (Figure 5B), some of the genera showed a similar changing trend between donor mice and recipient mice, as shown in Figure 5C,D. Among the high abundance genera, we found that the levels of *Allobaculum* and *Bacteroides* were increased, and the level of *Desulfovibrio* was decreased, after microbiota treatment in HFDR→HFD compared with HFD→HFD (Figure 5C). Among the low abundance genera, the level of *Parasutterella* was increased; and the levels of *Oscillibacter*, *Anaerotruncus*, *Ruminococcaceae_UCG-014* and *Coprococcus_1* were decreased in the HFDR→HFD group compared with the HFD→HFD group (Figure 5D). These trends were consistent with those obtained in the HFD-fed and HFDR donor groups (Figure 2C,D and Figure 5C,D).

## 4. Discussion

Obesity, chronic low-grade inflammation and metabolic syndrome are results of the mutual influences of the gut bacteria, host genome and diet [18]. RSV, the major active ingredient in grapes, peanuts and jackfruit, has been reported to improve disordered lipid metabolism, inflammation and other metabolic syndrome symptoms [19,20]. Recent evidence showed that gut bacterial dysbiosis plays a causal role in the development of chronic low-grade inflammation, an underlying factor of obesity and related health complications [21,22]. In this study, we proved that RSV treatment in HFD-fed mice improved systemic and metabolic tissue inflammation by regulating the gut microbiota’s structure. Notably, we observed that RSV partially prevents HFD-induced alterations in the gut microbiota’s structure. Moreover, the FMT experiment showed similar effects to those observed in the HFDR group, such as anti-inflammation and a particular microbiota-genera-changing trend. Thus, these results demonstrate that modulation of the gut bacteria induced by RSV treatment has a therapeutic effect on chronic-low grade inflammation in HFD-fed mice.

HFD-induced obesity is accompanied by chronic low-grade inflammation [23]. Previous evidence suggested that adipose tissue macrophage accumulation plays a pivotal role in the development of obesity [24]. Visceral obesity presents prominent triglyceride deposition in adipose tissues, which can further secrete pro-inflammation cytokines into circulation; in turn, the amplified inflammatory signal contributes to the forming of adiposity and insulin resistance [17]. Our results demonstrated that RSV treatment significantly downregulated the serum levels of TNF-α and IL-6. Moreover, the metabolic tissue inflammation was also ameliorated by RSV treatment in HFD-fed mice, such as the levels of IL-6, IL-1β and MCP-1 in jejunum and WAT tissue; the levels of TNF-α, IL-6 and IL-1β in BAT tissue; and the level of IL-6 in liver tissue. Some interventions, such as grape seed, ginger, Luffa cylindrica and *platycodon grandifloras*, have also been found to reduce inflammation in HFD-fed mice [11,25,26].

The gastrointestinal tract has been considered as a target tissue for the production of circulating inflammation with a HFD [8,27]. Consistent with our results, Patrice et al. showed that a HFD contributes to increased gut permeability and inflammation [23]. HFD-induced LPS over-production causes destabilization of gut barrier function. The disruption of gut permeability can further increase the circulating levels of LPS, which in turn cause systematic inflammation and metabolic disorders [17]. In our study, RSV treatment significantly decreased the serum absorbance of FITC–dextran and concentration of LPS content in the HFD group, which indicated repaired gut barrier function.

The beneficial effects of RSV on systematic inflammation and intestinal barrier integrity may regulated by the improved gut microbiota composition. Notably, we found that RSV improved the gut bacteria composition after it was degraded by the consumption of a HFD. Regarding gut bacteria composition on the genus level, RSV raised the prevalence of the high-abundance genera, such as *Allobaculum*, *Bacteroides* and *Blautia*, in the HFD group. Previous studies showed that *Allobaculum*, *Bacteroides* and *Blautia* genus produce SCFAs and were negatively correlated with inflammation, insulin resistance, obesity and other metabolic disorders [28,29,30,31]. Moreover, we observed the endotoxin-producing bacteria *Desulfovibrio* decreased in the gut microbiota of HFDR mice. The abundance of *Desulfovibrio* is commonly increased after a fat-enriched diet and can be decreased by probiotic-enriched diet intervention. Consistently, Emily et al. showed that *Desulfovibrio* flourish in an inflammatory environment [32]. Additionally, in the present study, we explored genera with low abundances but significant differences in their levels in the HFD and HFDR groups. We found that the abundance of *Parabacteroides* was increased in the HFDR group. *Parabacteroides* are highly effective commensal bacteria in the intestine. Kai et al. analyzed 736 American gut samples and found that the abundance of *Parabacteroides* had a significantly negative relationship with body mass index [33]. Furthermore, Kai et al. conducted in vivo experiments to validate the beneficial effects of *Parabacteroides distasonis* and showed that oral treatment with *Parabacteroides distasonis* reduced weight gain, improved glucose homeostasis and alleviated obesity-related abnormalities. Consistently, Wu et al. reported that oral treatment with *Parabacteroides goldsteinii* in HFD-fed mice reduced obesity and levels of inflammation. These evidences indicate that RSV and RSV-promoted gut commensal bacteria represent novel prebiotics and probiotic to treat obesity and inflammation [34].

Despite the potent anti-inflammatory effect of RSV in various in vivo and in vitro models, sometimes its anti-inflammation effect seems to be lacking in clinical studies. In one human study with nine healthy men and women, the participants were given 1 g/day of RSV for 28 days. The results showed that RSV reduced their plasma levels of TNF-α [35]. Another study in the administration of 75 mg/day of RSV over 12 weeks to 14 non-obese postmenopausal women showed no changes in inflammatory markers [36]. A possible reason for the contrasting results is the different doses of RSV. Due to the relatively low bioavailability of RSV, the concentration of RSV is very low in the serum after oral administration of low doses of RSV, thereby not facilitating significantly positive health effects. According to the metabolic characteristics of RSV, the intestine is a crucial site for its health benefits. The different proportions of bacteria and levels of metabolic activity of them in the intestine may be responsible for the individual variations in the health effects of RSV. Therefore, treatment of the microbiome directly has been emerging as an attractive therapy [15,33,37]. In the present study, FMT experiments revealed that RSV-altered gut bacteria have a causal role in the protection of systematic inflammation and host metabolism. Our results found that daily feeding with HFDR-indued microbiota decreased the pro-inflammatory cytokine levels in serum and metabolic tissue. Moreover, the 16S rRNA analyses showed similar results of gut bacteria composition for the recipient mice and donor mice. From the above-mentioned evidence, we proved that the RSV-mediated systematic inflammation improvements were, at least partially, associated with gut bacteria. More evidence is required to detect the gut microbiota that are functionally tied to RSV and their mechanisms of modulating systematic inflammation and host metabolism.

## 5. Conclusions

In summary, our results indicate that RSV treatment altered the structure of the gut bacteria, resulting in beneficial activity against systematic inflammation and obesity in HFD-fed mice. Further studies are needed to explore the causal link between the RSV-induced inflammation improvement and the key gut bacteria. Our study provides the basis for illumination of the underlying mechanisms of the health effects of RSV via mediating key gut microbiota.

## Figures and Tables

**Figure 1 nutrients-14-01994-f001:**
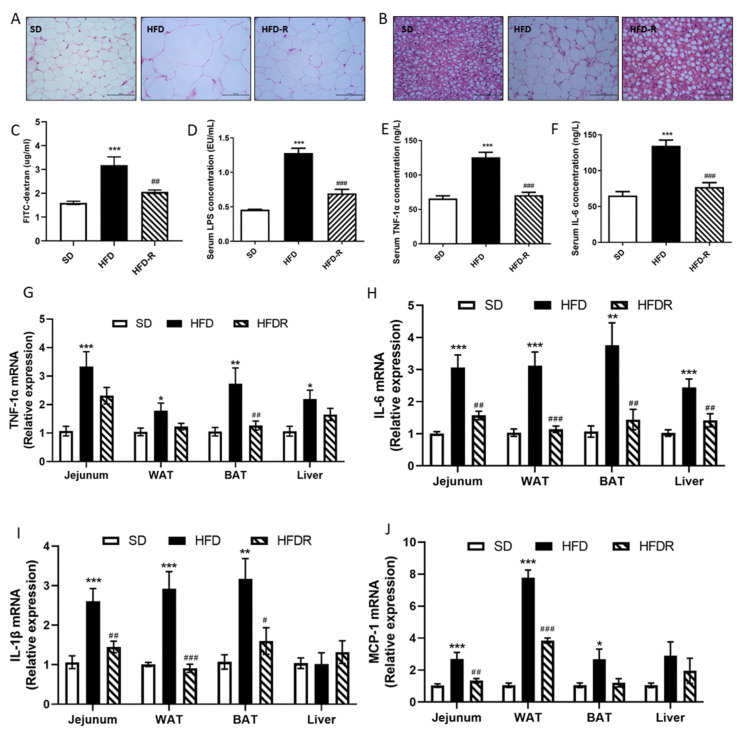
RSV reduced fat accumulation and systemic low-grade inflammation in a HFD group. (**A**) H&E staining of different WAT sections (scale: 50 μm); (**B**) H&E staining of different BAT sections (scale: 50 μm); (**C**) FITC–dextran translocation across the gut barrier into serum in the three groups; (**D**) concentration of serum LPS measured by a limulus amebocyte lysate assay kit; (**E**) serum tumor necrosis factor-α (TNFα) concentration; (**F**) serum IL-6 concentration; (**G**) the relative expression levels of TNF-1α in jejunum, WAT, BAT and liver tissue; (**H**) the relative expression levels of IL-6 in jejunum, WAT, BAT and liver tissue; (**I**) the relative expression levels of IL-1β in jejunum, WAT, BAT and liver tissue; (**J**) the relative expression levels of MCP-1 in jejunum, WAT, BAT and liver tissue. Values shown as the means ± SEM (*n* = 6). Differences were assessed by ANOVA and are denoted as follows: vs. SD * *p* < 0.05, ** *p* < 0.01, *** *p* < 0.001; vs. HFD ^#^ *p* < 0.05, ^##^ *p* < 0.01, ^###^ *p* < 0.001. WAT, epididymal white adipose tissue; BAT, interscapular brown adipose tissue.

**Figure 2 nutrients-14-01994-f002:**
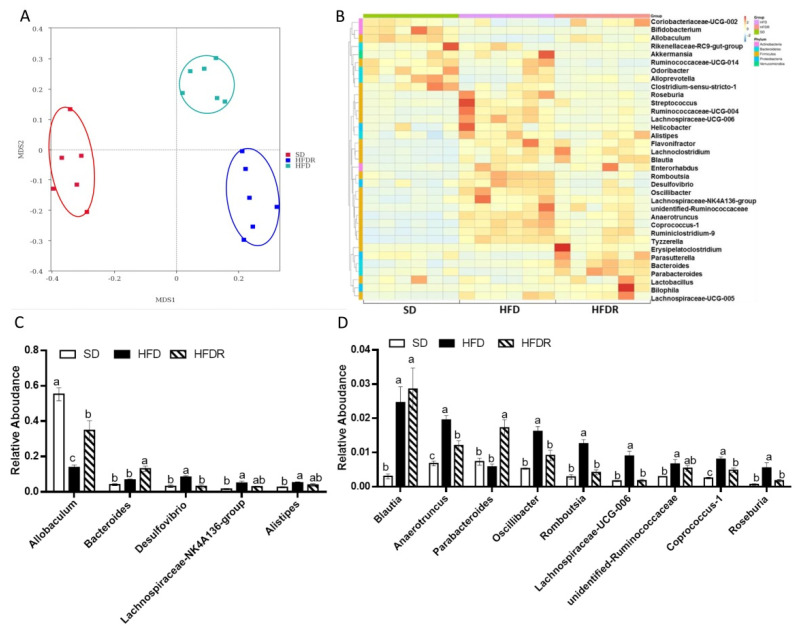
Dietary RSV improved the gut microbiota’s structure in HFD-fed mice. (**A**) Nonmetric multidimensional scaling (NMDS) score plot based on the Bray–Curtis plot; (**B**) Heatmap comparison and a hierarchical clustering dendrogram based on the relative abundances of the top 35 genera across all mice. (**C**) High-abundance genera, (**D**) low-abundance genera. The dots with different colors represent the important bacteria in each group with the same color. Values are shown as the means ± SEM (*n* = 10). Bars in (**C**,**D**) marked with different letters on top represent statistically significant results (*p* < 0.05) based on Tukey’s post hoc one-way ANOVA analysis, whereas bars labelled with the same letter correspond to results with no statistically significant differences. In cases where two letters are present on top of the bar in (**C**,**D**), each letter should be compared separately with the letters of other bars to determine whether the results show statistically significant differences.

**Figure 3 nutrients-14-01994-f003:**
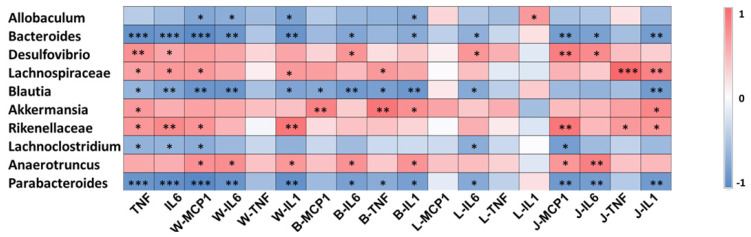
Gut microbiota involved in the systematic-inflammation-alleviating effects of RSV. Heatmap analysis of the Pearson correlations of gut microbiota and inflammation-related indexes. Statistical significance was determined by one-way ANOVA with Tukey tests for multiple-group comparisons. * *p* < 0.05, ** *p* < 0.01, and *** *p* < 0.001. Serum: TNF, IL-6; WAT: W-MCP1, W-IL6, W-TNF, W-IL1; BAT: B-MCP1, B-IL6, B-TNF, B-IL1; liver: L-MCP1, L-IL6, L-TNF, L-IL1; jejunum: J-MCP1, J-IL6, J-TNF, J-IL1.

**Figure 4 nutrients-14-01994-f004:**
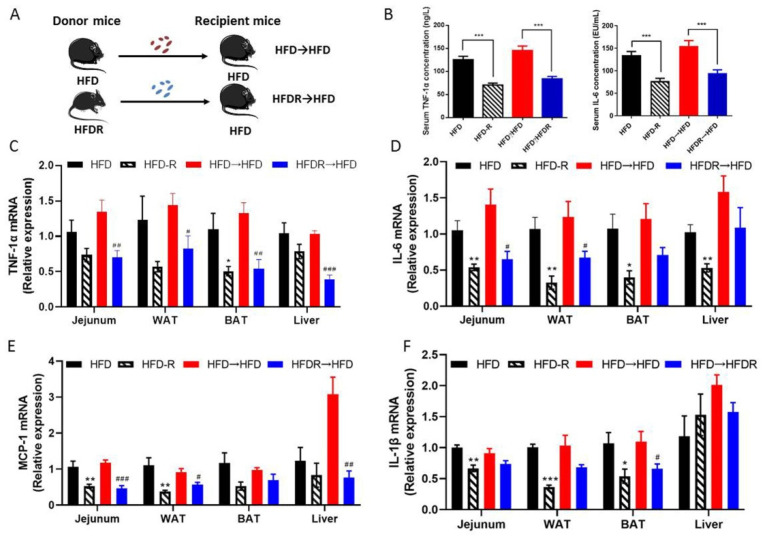
Effect of FMT on systematic inflammation. (**A**) Experiment design; (**B**) serum TNFα and IL-6 concentration; (**C**) the relative expression levels of TNF-1α in the jejunum, WAT, BAT and liver tissue; (**D**) the relative expression levels of IL-6 in the jejunum, WAT, BAT and liver tissue; (**E**) the relative expression levels of IL-1β in the jejunum, WAT, BAT and liver tissue; (**F**) the relative expression levels of MCP-1 in the jejunum, WAT, BAT and liver tissue. Data are expressed as the mean ± SEM (*n* = 6). Differences were assessed by unpaired Student’s *t* tests and are denoted as follows: HFDR vs. HFD * *p* < 0.05, ** *p* < 0.01, *** *p* < 0.001; HFDR→HFD vs. HFD→HFD ^#^ *p* < 0.05, ^##^ *p* < 0.01, ^###^ *p* < 0.001.

**Figure 5 nutrients-14-01994-f005:**
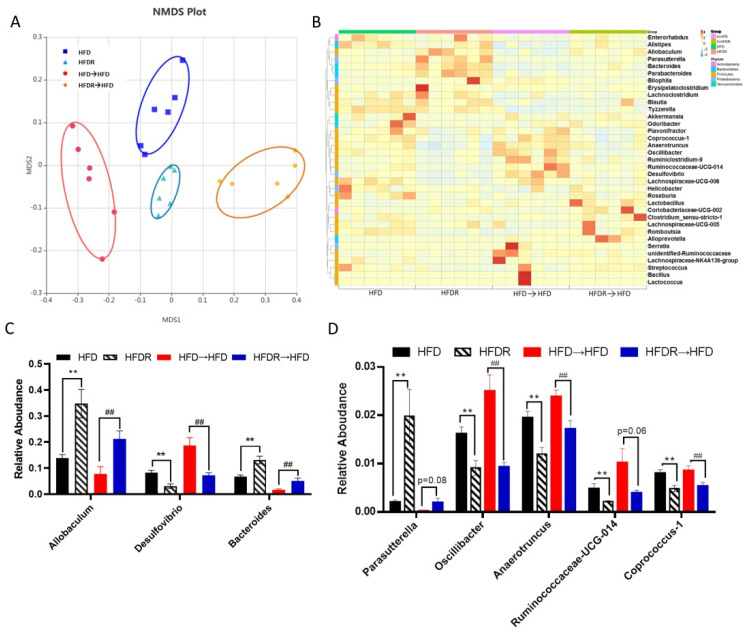
Effect of FMT on the gut microbiota’s structure. (**A**) Nonmetric multidimensional scaling (NMDS) score plot based on the Bray–Curtis plot. (**B**) Heatmap comparison and hierarchical clustering dendrogram based on the relative abundances of the top 35 genera across all mice. (**C**) High-abundance genera. (**D**) Low-abundance genera. Data are expressed as the means ± SEM (*n* = 6). Differences were assessed by based on unpaired Student’s *t* tests and are denoted as follows: vs. HFD ** *p* < 0.01; vs. HFD→HFD ^##^ *p* < 0.01.

## Data Availability

The number of the sequencing project is PRJNA827600.

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
