# Peer review of "Protective Effects of Dietary Resveratrol against Chronic Low-Grade Inflammation Mediated through the Gut Microbiota in High-Fat Diet Mice"

_nutrients, 2022, doi:10.3390/nu14101994_

Round 1
Reviewer 1 Report
Overall the study evidence changes in gut microbiota composition and the anti-inflammatory effect of RSV. However, some parts of the methods sections are unclear, mainly related to 16S rRNA gene analysis and OTU assignment, and have some statistical flaws.
Mainly I have the following concerns:
Methods:
- The number of subjects used to conduct the experiments is unclear in the methods section:
- Mice fed with SD n=12.
- Mice fed with HFD + RSV gavage n =12
- Mide fed with HFD n= Not clear
- Mice fed with HFD + FMT from HFD donors n= Not clear
- Mice fed with HFD + FMT from HFD-RSV donors n= 6?. Not clear also.
- A number indicating the subjects tested in that particular experiment is mentioned. Nevertheless, it is not clear if not all the samples were tested. In that case, what were the criteria used for selecting samples?i.e:
- 2.5. Gene expression analysis (LINE 119) n=6
- 2.7. 16S rRNA sequencing (LINE 126) n= 10.
- 2.2. Faecal transplantation (Lines 92 -97). This paragraph is not clear enough, and it is not easy to understand how many mice were in each group after FMT.
- 2.7. 16S rRNA sequencing (Lines 124-130). What was the methodology used for the OTU assignment, and what database was used?. How do the authors calculate alfa and beta diversity indexes? The authors used LEfSe, but it is not in the methods section, and it is not clear what LDA and alpha parameters were used.
- 2.8. Statistical analysis: The tests used assume that data is following a normal distribution. Do the authors test for normality? Microbiome data is usually not normally distributed, and a non-parametric test should be used. Also, for determining differences between beta diversity, a statistical analysis, either PERMANOVA or ANOSIM should be performed to determine the significance of differences between groups. In Figure 2 Authors said: "Differences were assessed with unpaired Student's t-tests," but in the methodology, they mentioned ANOVA. Please clarify and use the proper methods taking into account the number of samples and distribution of the sample.
Results & Discussion:
- Lines 141 -142: "To investigate the impact of RSV on obesity and systemic low-grade inflammation, RSV were supplied to mice were fed either a SD or HFD for 16 weeks". Comment: According to the previous statement, it seems like RSV was also supplemented to SD-fed mice; please clarify.
- Lines 230-232. Notably, the results also indicated that the levels of the two metabolites and RSV were positively correlated with systematic inflammation features. Comment: The previous sentence seems misplaced; the authors did not mention metabolites in other manuscript sections. To what metabolites do the authors refer. Please clarify.
- Figure 3. Gut microbiota and metabolites involved in systematic inflammation alleviating effects of RSV. Heatmap analysis of the pearson correlation of gut microbiota and inflammation-related indexes. Statistical significance was determined by one-way ANOVA with Tukey tests for multiple-group comparisons. *P < 0.05, **P < 0.01, and ***P < 0.001. Comment: Authors mention "metabolites," but it is unclear what metabolites they refer to. Also, in the figure, the name of the cytokines begins with "B-"," M-", or "G-"what does this mean? Please clarify.
- Lines 260-263: Upon transplantation, the recipient mice exhibit similar microbiota composition with their corresponding donor groups as shown in the PCoA and the NMDS results (Fig. 5A). Moreover, recipient HFD-fed mice showed altered microbiota composition after FMT from HFD-R-fed mice compared with mice that received stools from HFD group (Fig. 5A). Comment: To correctly state that the beta diversity of donors and recipients is similar, a statistical analysis is needed, either PERMANOVA or ANOSIM. Also, the quality of Figure 5A is not well.
- Lines 291 -293: We demonstrated that RSV efficiently suppressed HFD-induced systematic inflammation. Notably, we observed that RSV partially prevents HFD-induced alteration in the gut microbiota structure and increases the accumulation of the same aromatic metabolites. Comment: What aromatic metabolites are the authors referring to?
Minor comments:
- There are some typos around the manuscript: Firmicute, Akkermansie, structure instead of composition, indexed instead of indexes.
- Line 229-230 "positively – negatively" instead of negatively.
- 223-224: "10 significant altered gut microbiotas" instead of "10 differentially abundant genera".
Figures:
- Figure 2. Significance is denoted with letters, but the legend is explained with #. Please correct.
Supplementary material:
- Figure 1: Please correct the legend. Significant appears with letters, but the legend is explained in an asterisk.
- Figure 3: Titles of the bar plots and legends are misplaced. (A) Phylum, (B) Order, (C) Family.
- Figure 4: There is a typo in the y-axis. Statistics are missing.
- Table 1: There are 33 primers mentioned in the table. Are all of those genes analyzed? There is no information in the manuscript about all of them.
- Sup. Tables 3 to 6: There is no explanation of significance in the legends.
- Sup. Table 6. Explain the abbreviations "Epi-WAT, Per-WAT, Ing-WAT".
Author Response
Manuscript ID: nutrients-1668995
Type of manuscript: Article
Title: Protective effects of dietary Resveratrol against chronic-low grade inflammation Mediated through the Gut microbiota in high-fat diet mice
Pan Wang, Yue Ma, Dang Wang, Yubin Wang, Wenting Zhao, Yanyan Zheng, Yujia Xu, Xiaosong Hu, Fang Chen* and Xiaoyan Zhao*
Journal: Nutrients
Dear Dr. Maria Luz Fernandez and Dr. Lluis Serra-Majem,
We greatly appreciated the timely, rigorous and thoughtful reviews about our manuscript. Thanks for giving us the opportunity to revise our manuscript. We have carefully checked and revised the manuscript. In order to meet the constructive criticisms and suggestions of the Reviewers, we have revised the manuscript with “track changes” and new data added in response to the comments raised by the editor and reviewers. A point-by-point response to these comments is provided in the attached pages.
We hope that the revised version of the manuscript is now suitable for publication at Nutrients.
Sincerely yours,
Fang Chen, Ph.D
Xiaoyan Zhao, Ph.D
Comments from the editors and reviewers:
Reviewer 1
Overall, the study evidence changes in gut microbiota composition and the anti-inflammatory effect of RSV. However, some parts of the methods sections are unclear, mainly related to 16S rRNA gene analysis and OTU assignment, and have some statistical flaws.
Response: We would like to thank the reviewer for the overall comments for our manuscript. We have modified the Statistical analysis in Line152-161. And we have added the description of sequencing data analysis in Line 143-150 and added Supplementary method.
Mainly I have the following concerns:
Methods:
- The number of subjects used to conduct the experiments is unclear in the methods section:
- Mice fed with SD n=12.
- Mice fed with HFD + RSV gavage n =12
- Mide fed with HFD n= Not clear
- Mice fed with HFD + FMT from HFD donors n= Not clear
- Mice fed with HFD + FMT from HFD-RSV donors n= 6?. Not clear also.
Response: We appreciate the reviewer’s suggestion. We have checked the sample number we used in different experiment. We have modified these sentences in Line 93, 101, 107.
- A number indicating the subjects tested in that particular experiment is mentioned. Nevertheless, it is not clear if not all the samples were tested. In that case, what were the criteria used for selecting samples?i.e:
- 2.5. Gene expression analysis (LINE 119) n=6
- 2.7. 16S rRNA sequencing (LINE 126) n= 10.
Response: We appreciate the reviewer’s suggestion. We have checked the sample number we used in different experiment. We have added these mentions in Line 130, 143-145.
- 2. Faecal transplantation (Lines 92 -97). This paragraph is not clear enough, and it is not easy to understand how many mice were in each group after FMT.
Response: We appreciate the reviewer’s suggestion. We have modified this sentence in Line 100-113.
- 7. 16S rRNA sequencing (Lines 124-130). What was the methodology used for the OTU assignment, and what database was used?. How do the authors calculate alfa and beta diversity indexes? The authors used LEfSe, but it is not in the methods section, and it is not clear what LDA and alpha parameters were used.
Response: We appreciate the reviewer’s suggestion. And we have added the description of sequencing data analysis in Line 143-150 and added Supplementary method.
- 8. Statistical analysis: The tests used assume that data is following a normal distribution. Do the authors test for normality? Microbiome data is usually not normally distributed, and a non-parametric test should be used. Also, for determining differences between beta diversity, a statistical analysis, either PERMANOVA or ANOSIM should be performed to determine the significance of differences between groups. In Figure 2 Authors said: "Differences were assessed with unpaired Student's t-tests," but in the methodology, they mentioned ANOVA. Please clarify and use the proper methods taking into account the number of samples and distribution of the sample.
Response: We appreciate the reviewer’s suggestion. And we have added the description of Statistical analysis in Line 152-161. We have performed ANOSIM analysis between groups in Supplementary Figure 2.
Results & Discussion:
- Lines 141 -142: "To investigate the impact of RSV on obesity and systemic low-grade inflammation, RSV were supplied to mice were fed either a SD or HFD for 16 weeks". Comment: According to the previous statement, it seems like RSV was also supplemented to SD-fed mice; please clarify.
Response: We appreciate the reviewer’s suggestion and sorry about our carelessness. We have modified this sentence in Line 165-168.
- Lines 230-232. Notably, the results also indicated that the levels of the two metabolites and RSV were positively correlated with systematic inflammation features. Comment: The previous sentence seems misplaced; the authors did not mention metabolites in other manuscript sections. To what metabolites do the authors refer. Please clarify.
Response: We appreciate the reviewer’s suggestion and sorry about our carelessness. We have deleted this sentence in Line 277-279.
- Figure 3. Gut microbiota and metabolites involved in systematic inflammation alleviating effects of RSV. Heatmap analysis of the pearson correlation of gut microbiota and inflammation-related indexes. Statistical significance was determined by one-way ANOVA with Tukey tests for multiple-group comparisons. *P < 0.05, **P < 0.01, and ***P < 0.001. Comment: Authors mention "metabolites," but it is unclear what metabolites they refer to. Also, in the figure, the name of the cytokines begins with "B-"," M-", or "G-"what does this mean? Please clarify.
Response: We appreciate the reviewer’s suggestion and sorry about our carelessness. We have modified this sentence in Line 281 and explained the abbreviation in Line 284-286.
- Lines 260-263: Upon transplantation, the recipient mice exhibit similar microbiota composition with their corresponding donor groups as shown in the PCoA and the NMDS results (Fig. 5A). Moreover, recipient HFD-fed mice showed altered microbiota composition after FMT from HFD-R-fed mice compared with mice that received stools from HFD group (Fig. 5A). Comment: To correctly state that the beta diversity of donors and recipients is similar, a statistical analysis is needed, either PERMANOVA or ANOSIM. Also, the quality of Figure 5A is not well.
Response: We appreciate the reviewer’s suggestion. It’s a good question. We have done the ANOSIM analyses and found that the beta diversity was different among the four groups (Supplementary Fig.5). So, we modified the description in Line 310-330.
- Lines 291 -293: We demonstrated that RSV efficiently suppressed HFD-induced systematic inflammation. Notably, we observed that RSV partially prevents HFD-induced alteration in the gut microbiota structure and increases the accumulation of the same aromatic metabolites. Comment: What aromatic metabolites are the authors referring to?
Response: We appreciate the reviewer’s suggestion and sorry about our carelessness. We have modified this sentence in Line 353.
Minor comments:
- There are some typos around the manuscript: Firmicute, Akkermansie, structure instead of composition, indexed instead of indexes.
- Line 229-230 "positively – negatively" instead of negatively.
- 223-224: "10 significant altered gut microbiotas" instead of "10 differentially abundant genera".
Response: We appreciate the reviewer’s suggestion and sorry about our carelessness. We have modified these typos in the whole manuscript.
Figures:
- Figure 2. Significance is denoted with letters, but the legend is explained with #. Please correct.
Response: We appreciate the reviewer’s suggestion. We have modified Figure 2 in the revised manuscript.
Supplementary material:
- Figure 1: Please correct the legend. Significant appears with letters, but the legend is explained in an asterisk.
Response: We appreciate the reviewer’s suggestion. We have modified the supplementary figure 1 legend in the revised supplementary material.
- Figure 3: Titles of the bar plots and legends are misplaced. (A) Phylum, (B) Order, (C) Family.
Response: We appreciate the reviewer’s suggestion. We have modified the supplementary figure 3 legend in the revised supplementary material.
- Figure 4: There is a typo in the y-axis. Statistics are missing.
Response: We appreciate the reviewer’s suggestion. We have deleted the original supplementary Figure 4 and we have re-analyzed the data and added in figure 6D.
- Table 1: There are 33 primers mentioned in the table. Are all of those genes analyzed? There is no information in the manuscript about all of them.
Response: We appreciate the reviewer’s suggestion and sorry about our carelessness. We have modified the primers used in this study in the Table 2.
- Tables 3 to 6: There is no explanation of significance in the legends.
Response: We appreciate the reviewer’s suggestion. We have added the description of the significance in these legends.
- Table 6. Explain the abbreviations "Epi-WAT, Per-WAT, Ing-WAT".
Response: We appreciate the reviewer’s suggestion. We have explained these abbreviations in the table legend.
Reviewer 2 Report
Wand P. Et al present a study concerning the impact of resveratrol on the host via microbiota. The manuscript seems to be not proofread. It contains a high number of all kinds of mistakes: grammar, missing words, mixing different tenses in one sentence, stylistic mistakes… The materials and method as well as discussion sections are not sufficiently developed. Most of all, the authors already published at least two articles (two available in Pubmed PMID: 32305646 and 30718820) concerning resveratrol and microbiota. The articles are even cited in the current manuscript but without mentioning that they originate from the same group or describing the results. I assume that it is for the reason that one of the articles contains the same set of experiments and overlapping data as the currently reviewed article.
Minor
The figure should not span over 2 pages
The authors do not explain why and how they chose the two types of bacteria for the separate graphs.
For some statistically significant differences between the bacteria mentioned in the text, the authors do not show any figures/table/proof.
More details concerning statistical analysis are required. There is absolutely no description of sequencing data analysis.
The number of figures for sequencing data should be limited to the important ones and the rest should be moved to supplementary figures.
The legend for figure 2 does not explain what a, b, c in the figures stand for but it mentions # which does not appear in the figure.
There is no description of the generation of inflammation-related factors used to generate figure 3. It is also not mentioned what type of tissue was used for it.
Lines: 342-344 this reference does not seem to have anything in common with the presented manuscript. If it does, the authors should make it clear.
What do the authors mean by “drugging the microbiome”?
The authors should discuss the high dose of resveratrol applied in the study in the context of naturally occurring levels of resveratrol.
The discussion is very one-sided and considering the richness of the literature does not give sufficient context. Also, controversy concerning the lack of impact or negative results of resveratrol supplementation in human studies could be mentioned.
Please correct the typos and mistakes (up to 3 mistakes per line) in lines: 19, 36, 37, 47, 48, 97, 122, 125, 132, 135, 140, 141, 142, 144, 163, 217, 224, 314, 321, 323…
Author Response
Manuscript ID: nutrients-1668995
Type of manuscript: Article
Title: Protective effects of dietary Resveratrol against chronic-low grade inflammation Mediated through the Gut microbiota in high-fat diet mice
Pan Wang, Yue Ma, Dang Wang, Yubin Wang, Wenting Zhao, Yanyan Zheng, Yujia Xu, Xiaosong Hu, Fang Chen* and Xiaoyan Zhao*
Journal: Nutrients
Dear Dr. Maria Luz Fernandez and Dr. Lluis Serra-Majem,
We greatly appreciated the timely, rigorous and thoughtful reviews about our manuscript. Thanks for giving us the opportunity to revise our manuscript. We have carefully checked and revised the manuscript. In order to meet the constructive criticisms and suggestions of the Reviewers, we have revised the manuscript with “track changes” and new data added in response to the comments raised by the editor and reviewers. A point-by-point response to these comments is provided in the attached pages.
We hope that the revised version of the manuscript is now suitable for publication at Nutrients.
Sincerely yours,
Fang Chen, Ph.D
Xiaoyan Zhao, Ph.D
Comments from the editors and reviewers:
Reviewer 2
Wand P. Et al present a study concerning the impact of resveratrol on the host via microbiota. The manuscript seems to be not proofread. It contains a high number of all kinds of mistakes: grammar, missing words, mixing different tenses in one sentence, stylistic mistakes… The materials and method as well as discussion sections are not sufficiently developed. Most of all, the authors already published at least two articles (two available in Pubmed PMID: 32305646 and 30718820) concerning resveratrol and microbiota. The articles are even cited in the current manuscript but without mentioning that they originate from the same group or describing the results. I assume that it is for the reason that one of the articles contains the same set of experiments and overlapping data as the currently reviewed article.
Response: We would like to thank the reviewer for the overall comments for our manuscript and sorry about my carelessness. We have revised the mistakes in the whole manuscript.
Moreover, we appreciate the reviewer’s No.2 suggestion. It’s a good question. In fact, the PMID: 32305646 and the present study are from one big project (PMID:30718820 from a different project). The previous paper (PMID: 32305646) focused on the top 10 genera. But in the current study, we aim to explore some new genera that with low abundance but have significant differences between HFD and HFDR groups. In addition, the purpose of the previous paper (PMID: 32305646) was to investigate the relationship between obesity and gut microbiota. As you know, inflammation plays a key role in the development of obesity, so we further study the role of gut microbiota induced by Resveratrol against chronic-low grade inflammation in high-fat diet mice. And we have added sentences (Line 66-71, 225-231, 243-250) about the difference between the previous paper and the present paper. Thanks again for your good suggestion.
Minor
- The figure should not span over 2 pages
Response: We appreciate the reviewer’s suggestion. We have modified Figure 2.
- The authors do not explain why and how they chose the two types of bacteria for the separate graphs.
Response: We appreciate the reviewer’s suggestion. We have modified Figure 2 and other figures were moved to supplementary figure 2. We have explained why we chose these bacteria for the separate graphs in the figure legend.
- For some statistically significant differences between the bacteria mentioned in the text, the authors do not show any figures/table/proof.
Response: We appreciate the reviewer’s suggestion. We have added the related Figures in the whole manuscript.
- More details concerning statistical analysis are required. There is absolutely no description of sequencing data analysis.
Response: We appreciate the reviewer’s suggestion. We have modified the Statistical analysis in Line152-161. And we have added the description of sequencing data analysis in Line 143-150 and added the Supplementary method.
- The number of figures for sequencing data should be limited to the important ones and the rest should be moved to supplementary figures.
Response: We appreciate the reviewer’s suggestion. We have modified Figure 2 and other figures were moved to supplementary figure 2.
- The legend for figure 2 does not explain what a, b, c in the figures stand for but it mentions # which does not appear in the figure.
Response: We appreciate the reviewer’s suggestion. We have modified Figure 2 and other figures were moved to supplementary figure 2. We have explained the meaning of “a, b, c” used in the figures in the figure legend.
- There is no description of the generation of inflammation-related factors used to generate figure 3. It is also not mentioned what type of tissue was used for it. Response: We appreciate the reviewer’s suggestion. We have added the tissue type that was used in the correlation analysis in figure 3 legend.
- Lines: 342-344 this reference does not seem to have anything in common with the presented manuscript. If it does, the authors should make it clear.
Response: We appreciate the reviewer’s suggestion. We have deleted this reference in Line 433-434.
- What do the authors mean by “drugging the microbiome”?
Response: We appreciate the reviewer’s suggestion. We have modified this sentence in Line 426.
- The authors should discuss the high dose of resveratrol applied in the study in the context of naturally occurring levels of resveratrol.
Response: We appreciate the reviewer’s suggestion. After searching the reference about “resveratrol” and “gut microbiota”, we found that the dose of the resveratrol ranged from 100-400 mg/kg/day in most research [1-5]. Two studies use the dose of 15 mg/kg/day [6, 7]. Just to be safe though, we choose to use 300 mg/kg/day of RSV. In the future, we will explore the effect of natural dose of RSV on the gut microbiota structure.
[1] Qiao Y, Sun J, Xia S, Tang X, Shi Y, Le G. Effects of resveratrol on gut microbiota and fat storage in a mouse model with high-fat-induced obesity. Food & function. 2014;5:1241-9.
[2] Jung MJ, Lee J, Shin NR, Kim MS, Hyun DW, Yun JH, et al. Chronic Repression of mTOR Complex 2 Induces Changes in the Gut Microbiota of Diet-induced Obese Mice. Scientific reports. 2016;6:30887.
[3] Yang C, Deng Q, Xu J, Wang X, Hu C, Tang H, et al. Sinapic acid and resveratrol alleviate oxidative stress with modulation of gut microbiota in high-fat diet-fed rats. Food research international. 2019;116:1202-11.
[4] Alrafas HR, Busbee PB, Nagarkatti M, Nagarkatti PS. Resveratrol modulates the gut microbiota to prevent murine colitis development through induction of Tregs and suppression of Th17 cells. Journal of leukocyte biology. 2019.
[5] Suárez M, Ardid-Ruiz A, Ibars M, Mena P, del Rio D, Bladé C, et al. UNDERSTANDING THE ANTI-OBESITY EFFECTS OF RESVERATROL: PHASE-II CONJUGATES VS GUT MICROBIOTA METABOLITES. Scripta Scientifica Pharmaceutica. 2017;4.
[6] Eteberria U, Arias N, Boque N, Macarulla MT, Portillo MP, Martinez JA, et al. Reshaping faecal gut microbiota composition by the intake of trans-resveratrol and quercetin in high-fat sucrose diet-fed rats. Journal of Nutritional Biochemistry. 2015;26:651-60.
[7] Zhao L, Zhang Q, Ma W, Tian F, Shen H, Zhou M. A combination of quercetin and resveratrol reduces obesity in high-fat diet-fed rats by modulation of gut microbiota. Food & function. 2017;8:4644-56.
- The discussion is very one-sided and considering the richness of the literature does not give sufficient context. Also, controversy concerning the lack of impact or negative results of resveratrol supplementation in human studies could be mentioned.
Response: We appreciate the reviewer’s suggestion. We have modified the discussion and added a sentence in Line390-406, 415-426.
- Please correct the typos and mistakes (up to 3 mistakes per line) in lines: 19, 36, 37, 47, 48, 97, 122, 125, 132, 135, 140, 141, 142, 144, 163, 217, 224, 314, 321, 323…
Response: We appreciate the reviewer’s suggestion and sorry about my carelessness. We have revised these mistakes in the whole manuscript.
Round 2
Reviewer 2 Report
Thank you for applying all changes. Most of the issues have been solved; however, the manuscript still contains language mistakes. Please review the text carfuly.
Some of the issues to be corrected:
Please correct the sentence to make it clearer:
Moreover, the FMT experiment showed showed similar microbiota genera changing trend to microbiota-modulating effects similar to the oral RSV-feeding experiment.
Line 24: change inflammation to inflammatory.
Line 70: Please correct: Nevertheless, it remains unclear whether RSV could alleviate system inflammatory status by is mediated by the modulating gut microbiota in HFD-fed mice remains unclear.
Line 165: please change play to plays
Line 421: Please correct all mistakes: Due to the relative low bioavailability of RSV, low doses of RSV do not have enough free RSV in the circulation, thus do not have positive health effects.
Please change bacteria structure to bacteria composition
